# Secondary Metabolites with Anti-Inflammatory from the Roots of *Cimicifuga taiwanensis*

**DOI:** 10.3390/molecules27051657

**Published:** 2022-03-02

**Authors:** Jih-Jung Chen, Ming-Jen Cheng, Tzong-Huei Lee, Yueh-Hsiung Kuo, Chao-Tsen Lu

**Affiliations:** 1Department of Pharmacy, School of Pharmaceutical Sciences, National Yang Ming Chiao Tung University (NYCU), Taipei 112, Taiwan; jjungchen@nycu.edu.tw; 2Department of Medical Research, China Medical University Hospital, China Medical University, Taichung 404, Taiwan; 3Bioresource Collection and Research Center (BCRC), Food Industry Research and Development Institute (FIRDI), Hsinchu 300, Taiwan; chengfirdi@gmail.com; 4Institute of Fisheries Science, National Taiwan University, Taipei 10617, Taiwan; thlee1@ntu.edu.tw; 5Department of Chemistry, National Taiwan University, Taipei 106, Taiwan; luct9507@gmail.com; 6Department of Biotechnology, Asia University, Taichung 413, Taiwan; 7Department of Chinese Pharmaceutical Sciences and Chinese Medicine Resources, College of Pharmacy, China Medical University, Taichung 404, Taiwan; 8ChineseMedicine Research Center, ChinaMedical University, Taichung 404, Taiwan

**Keywords:** *Cimicifuga taiwanensis*, Ranunculaceae, cycloartane, antiinflammatory activities

## Abstract

The genus *Cimicifuga* is one of the smallest genera in the family Ranunculaceae. Cimicifugae Rhizoma originated from rhizomes of *Cimicifuga* simplex, and *C. dahurica*, *C. racemosa*, *C. foetida*, and *C. heracleifolia* have been used as anti-inflammatory, analgesic and antipyretic remedies in Chinese traditional medicine. Inflammation is related to many diseases. *Cimicifuga taiwanensis* was often used in folk therapy in Taiwan for inflammation. Phytochemical investigation and chromatographic separation of extracts from the roots of *Cimicifuga taiwanensis* has led to the isolation of six new compounds: cimicitaiwanins A–F (**1**–**6**, respectively). The structures of the new compounds were unambiguously elucidated on the basis of extensive spectroscopic data analysis (1D- and 2D-NMR, MS, and UV) and comparison with the literature data. The effect of some isolates on the inhibition of NO production in lipopolysaccharide-activated RAW 264.7 murine macrophages was evaluated. Of the isolates, **3**–**6** exhibited potent anti-NO production activity, with IC_50_ values ranging from 6.54 to 24.58 μM, respectively, compared with that of quercetin, an iNOS inhibitor with an IC_50_ value of 34.58 μM. This is the first report on metabolite from the endemic Taiwanese plant-*C. taiwanensis*.

## 1. Introduction

Inflammation is a self-protective mechanism designed to eliminate harmful stimuli, including damaged cells, irritants, or pathogens, and initiate the wound repair process. However, inflammation sometimes induces further inflammation, leading to self-perpetuating chronic inflammation that can lead to severe cellular and tissue damage [1]. Chronic inflammation is associated with a variety of diseases, such as atherosclerosis [2], Alzheimer’s disease [3], diabetes [4], and carcinogenesis [5].

Nitric oxide (NO) is a mediator in the inflammatory response involved in host defense. It is mainly produced by inducible nitric oxide synthase (iNOS) in inflammatory states [6] and plays a key role in every pathological process of the inflammatory process [7,8,9,10]. In a series of studies on the anti-inflammatory constituents of Formosan endemic plants, over 500 species were screened for the inhibitory activity on LPS-induced NO release in RAW 264.7 murine macrophages, and *Cimicifuga taiwanensis* was one of the active species.

The genus *Cimicifuga* has been widely used around the world and has been a traditional Chinese herbal medicine since ancient times [11]. There are 18 species of Cimicifugae Rhizoma in the temperate zone of the northern hemisphere, 8 of which are distributed in mainland China and only one specie, *C. taiwanensis* (J. Compton, Hedd., and T.Y.A. Yang) Luferov, in Taiwan [12,13]. Three *Cimicifuga* species, namely *C. dahurica*, *C. foetida*, and *C. heracleifolia*, commonly known as “Shengma” in Chinese, are officially listed in the Chinese Pharmacopoeia [14]. In traditional Chinese medicine, *Cimicifuga* is widely used for treating aphtha, sore throat, toothache, and wind heat headache. It also has been used in archoptosis, non-erupting measles, spot poison, uterine prolapse, and other diseases [14,15].

*C. taiwanensis* is a plant endemic to Taiwan and distributed in open areas or forest margins at an altitude of 2200–3200 m in the central mountainous area. To date, more than 450 compounds, including 9,19-cycloartane triterpenoids, phenylpropanoids, chromones, lignans, amides, and other compounds, have been isolated from *Cimicifuga* spp. plants [16,17,18,19,20]. Among them, many compounds such as triterpenoids [21,22,23,24] and phenolic compounds [25,26] show potent anti-cancer activities. There have been few studies on the anti-inflammatory active components of this genus in the past. We chose it for research based on the results of the primary screening activity and the reason that the phytochemical components of this plant have not been studied. This motivated us to study its bioactive metabolites. Investigation of the bioactive metabolites of the active MeOH extract from *C. taiwanensis* led to the isolation of six new compounds, namely cimicitaiwanins A–F (**1**‒**6**) (Figure 1). Their structures were characterized by means of spectroscopic methods, especially by ^1^H-, ^13^C-, and 2D-NMR (Figure 2 and Figure 3, Table 1, Table 2 and Table 3), as well as by HR-ESI-MS experiments and comparison with the literature data. The structural elucidation of the new isolates and the anti-inflammatory activities of the isolates are described herein.

## 2. Results and Discussion

### 2.1. Structure Elucidation of Compounds

Compound **1** was obtained as white needles and gave a molecular formula of C_30_H_46_O_6_, as determined by HR-EIMS ([M + Na]^+^ *m*/*z* 502.3303, calcd 502.3289), requiring eight degrees of unsaturation. The IR spectrum showed absorptions for the hydroxyl group at 3440 cm^−1^ and carbonyl group at 1711 cm^−1^. The ^1^H NMR spectrum (Table 1) showed the presence of the characteristic 9,19-cycloartane-type triterpene 7 methyl signals at δ_H_ 0.83, 0.92, 0.94, 0.99, 1.22, 1.32 (each 3H, s), and 1.08 (3H, d, *J* = 6.4 Hz), and 4 oxymethines appeared at δ_H_ 3.54 (1H, d, *J* = 4.0 Hz, H-24), 3.72 (1H, d, *J* = 5.6 Hz, H-3), 4.02 (1H, s, H-15), and 4.42 (1H, ddd, *J* = 9.6, 4.0, 3.2 Hz, H-23), while 1 cyclopropane methylene signal was found at δ_H_ 1.69 and 3.14 (each 1H, br d, *J* = 13.6 Hz). The ^13^C-DEPT spectra (Table 2) exhibited 30 carbon signals consisting of 7 methyls, 7 methylenes, 8 methine carbons, and 8 quaternary carbons (including 2 double bonds). The above-mentioned evidence suggested **1** as a derivative of the known 24-epi-acerinol [27]. The ^1^H (Table 1) and ^13^C NMR data of **1** were very similar to those of 24-epi-acerinol [27], except that an oxymethine group [δ_H_ 4.10 (1H, t, *J* = 7.6 Hz)] at C-12 in **1** replaced the methylene group of 24-epi-acerinol [27]. This was supported by NOESY correlations between H-12 (δ_H_ 4.10)/ CH_3_-28 (δ_H_ 0.92) and by HMBC correlation between H-12 (δ_H_ 4.10) and C-13 (δ_C_ 46.0) (Figure 3). In addition, the signal of CH_3_-21 shifted from δ_H_ 0.88 to 1.08 ppm in a low magnetic field, which also showed that it was affected by the hydroxyl group substituted by 12β (similar position in space), so **1** was determined to be 24-epi-12β-hydroxyacerinol and named cimicitaiwanin A.

Compound **2** was obtained as white needles. The HR-EIMS of **2** gave an [M]^+^ ion peak at *m*/*z* 544.3377 (calcd 544.3395), consistent with the molecular formula C_32_H_48_O_7_, indicating nine degrees of unsaturation. The IR spectrum showed absorptions for the hydroxyl group at 3452 cm^−1^, acetoxy (1248 and 1737 cm^−1^), and a double bond at 1651 cm^−1^. The ^13^C-DEPT spectra exhibited 32 carbon signals (Table 2), classified as 8 methyls, 7 methylenes, 8 methines (3 oxygenated), and 9 quaternary carbons (3 oxygenated, 1 olefinic, 1 olefinic, and 1 acetal group). The diagnostic signals of 2 oxygen-bearing methine carbons at δ_C_ 86.0 (C-24) and δ_C_ 71.8 (C-23) and an acetal quaternary carbon at δ_C_ 111.9 suggested **2** was also a cimigenol-type triterpene. Upon further inspection of the ^1^H NMR and HSQC spectrums of **2**, the NMR data (Table 1 and Table 2) of **2** was very similar to those of **1**, and the difference between the two compounds was that in addition to the different splitting forms of H-24 on the ^1^H-NMR, which means that the relative configuration of C-24 was different, there should have been other signals present in the molecule from δ_H_ 3.54 to 3.90 in the lower magnetic field, and CH_3_-26 and 27 are also shifted to the lower magnetic field by ~0.2 ppm, indicating that there should be a strong electron-withdrawing group nearby, so it was speculated that the acetyloxy group was substituted into the position of C-25. This inference could not be confirmed from two-dimensional experiments, but C-25 also shifted from δ_C_ 68.7 to 82.3 downfield, which was consistent with this inference. Therefore, the structure of **2** was determined to be 25-O-acetyl-12β-hydroxyacerinol and given the name cimicitaiwanin B.

Compound **3** was a white needle-like crystal with a melting point of 103–105 °C, where 
$\alpha_{D}^{26}$
 = +10.07 (c 0.29, CHCl_3_). The HREIMS (*m*/*z* 588.3185, [M]^+^; calculated for C_32_H_46_O_8_, 588.3187) of compound **3** suggested a molecular formula of C_32_H_46_O_8_, with 11 degrees of unsaturation. The UV maxima absorption of **3** at 255 nm indicated the presence of one double bond (C=C)-conjugated carbonyl group. IR showed the presence of conjugated carbonyl groups (1661 cm^−1^), acetoxyl groups (1249 and 1732 cm^−1^), and hydroxyl groups (3419 cm^−1^). The presence of one conjugated carbonyl group and acetoxy were revealed by IR absorptions, along with the resonance signals in the NMR spectra (δ_C_ = 198.5 (C=O-C=C, C-11); δ_C_ = 170.4 (OCOMe), 22.5 (OCOMe); δ_H_ = 1.99 ((OCOMe)). After deducting these two carbons, there were still 30 left, and then we observed that the ^1^H-NMR spectrum displayed 2 7-membered rings of cycloartane C-19 methylene at δ_H_ 2.69 and 3.15 (each 1H, br d, *J* = 13.6 Hz), 6 singlet methyl groups at δ_H_ 0.98, 0.99, 1.05, 1.15, 1.41, and 1.46 (each 3H), a doublet methyl group at δ_H_ 0.91 (3H, d, *J* = 6.8 Hz), and 4 oxygenated methines at δ_H_ 3.80 (1H, d, *J* = 5.2 Hz, H-3), 4.79 (1H, dd, *J* = 10.2, 7.6 Hz, H-7), 4.40 (1H, d, *J* = 9.2 Hz, H-23), and 3.93 (1H, s, H-24), showing that **3** was a 7-membered cyclic oxysubstituted cycloartane. In addition, the two double-bonded carbon signals on the seven-membered ring cycloartane skeleton at δ_C_ 130.1 and 158.9 were lower than the usual magnetic field, indicating that the carbonyl group was conjugated with this C-8/C-9 tetra-substituted double bond, and the position was a product of the HMBC correlations between C-11/H-12 and C-12/Me-18, confirming its position at C-11. Because of the closeness in space, the displacement of H-1β and H-19α to the lower magnetic field also confirmed that the carbonyl group was substituted on C-11.

One low magnetic field of oxymethine appeared at δ_H_ 4.79 (dd, *J* = 10.2, 7.2 Hz, H-7), which was presumed to be replaced at the allylic position. The conjecture was confirmed by HMBC correlation between C-8/H-7. In addition, because H-7 had a large split with one of H-6 at 10.2 Hz, it was speculated that H-7 was located in the axial direction, which was arranged in the α orientation by the model. This could be confirmed by the NOESY correlation of H-7/H-5. The signal of H-15 was shifted to δ_H_ 4.74 ppm in the lower magnetic field rather than the usual δ_H_ 4.10, which should have been affected by the sterically adjacent hydroxyl groups. This could also be used to confirm that the substitution was indeed in the 7β position. The positions of the acetoxy groups were shifted downfield from the signals of H-24, CH_3_-26, CH_3_-27, and C-25, presumably as **2** was on C-25. Therefore, **3** was identified as 25-O-acetyl-7β-hydroxy-11-oxoacerinol and was named cimicitaiwanin C.

Compound **4** was a yellow-green oily substance with a molecular ion peak at *m*/*z* 660 in EI-MS with 
$\alpha_{D}^{21}$
 = +9.76 (c 0.75, CHCl_3_). IR showed the presence of double bonds (1665 cm^−1^), acetyloxy (1251 and 1731 cm^−1^), and hydroxyl groups (3426 cm^−1^). The ^13^C-NMR spectrum showed that **4** had 37 carbons, including 11 oxygen carbons, of which 2 were connected to 2 oxygens located at δ_C_ 104.1 (C-1′) and 112.0 (C-16). The latter was the C-16 signal of a typical polyoxy-substituted polycyclic cycloartane, and the former was a typical anomeric C-1 signal. The H-19 methylene group of the seven-membered ring cycloartane and the signals of many coupled oxymethines were also observed in the ^1^H-NMR spectrum. It was speculated that **4** was a triterpenoid glycoside. Therefore, of the 37 carbons, 30 were cycloartane, 2 carbons were an acetyloxy group that could be easily seen from the ^1^H and ^13^C spectra, and the remaining 5 carbons showed that this sugar was a 5-carbon sugar.

According to the principle of the same coupling constant, six protons on the oxygenated carbons that were coupled to each other could be found. The relationship between C-1’/H-3 and H-3’ on sugar could be revealed by the HMBC contacts, and the position of H-1’ could be determined accordingly. Then, there was the connection of H-4’ and two protons (CH_2_-5’) on the oxygen-containing secondary carbon with C-3’. The secondary carbon (CH_2_-5) was then correlated to H-1’ in HMBC, which could be associated with a complete xylose. If cycloartane was placed in equatorially to this sugar, H-1’ was axial. It can be seen from NOESY that H-1’ was correlated to H-3’ and Hα-5’, confirming that the hydroxyl group was substituted at the β position of C-3’. H-2’, and H-4’, which could be correlated to each other, showing that the hydroxyl groups of C-2’ and C-4’ were both placed equatorially (i.e., the α position). We determined the relative configuration of each carbon on the sugar, compared it with the literature [28], and determined it to be xylose.

The remaining 4 protons on the oxygenated carbon were the same as those of **1**, except that H-3 is shifted to the lower magnetic field by ~0.1 ppm due to the sugar connection. It has been speculated that the D ring and the side chain of compound **4** is the same as that of **1**. The ^13^C-NMR spectrum showed four double bonds (δ_C_ 116.9, 126.8, 136.3, and 137.8), but only one olefin proton appeared at δ_H_ 5.20 (1H, br s), and DEPT confirmed that one of the two double bonds was tri-substituted, while another one was tetra-substituted. The 2D-NMR experiments confirmed the double bond substitution between C-1/C-10 and C-8/C-9. Then, CH_2_-19 was located at the allylic position of the two double bonds, and the signal was lower in the magnetic field than the CH_2_-19 position of the general seven-membered ring cycloartane, located at δ_H_ 2.41 and 2.83 (each 1H, d, *J* = 14.8 Hz). As mentioned for the previous compounds, the position of the acetyloxy group could be easily determined by NMR. Consequently, compound **4** was elucidated as cimicitaiwanin D.

The HREI-MS (*m*/*z* 482.3027, [M]^+^; calcd for C_30_H_42_O_5_, 486.3027) of compound **5** suggested a molecular formula of C_30_H_42_O_5_ with eight degrees of unsaturation. Its IR spectrum showed hydroxyl absorptions at 3508 cm^−1^ and a double bond at 1668 and 1614 cm^−1^. The ^1^H and ^13^C NMR data (Table 1 and Table 2) for **5** were similar to those of cimigenol-1(2),7(8)-dien-3-one [29]. The main differences between them were the S-configuration on H-24 in cimigenol-1(2),7(8)-dien-3-one [28] (H-23 (δ_H_ 4.77 [t, *J* = 8.0 Hz]) and H-24 (δ_H_ 3.80 [0 Hz])) being replaced by the R-form by comparison of the coupling constants of H-23 (δ_H_ 4.46 (br d, *J* = 9.8 Hz)) and H-24 (δ_H_ 3.57 (t, *J* = 4.4 Hz)) with those of the known ones. Therefore, the structure of **5** was determined as shown and given the name cimicitaiwanin E.

Compound **6** was obtained as an optically active oil (
$\alpha_{D}^{26}$
 = −9.70 (c 0.27, CHCl_3_)). The molecular formula of **6** was determined to be C_15_H_24_O_3_ from the HR-EI-MS data (*m*/*z* 252.1727 ([M]^+^; calculated to be 252.1720)) as well as from its ^13^C-NMR and DEPT, requiring four degrees of unsaturation. The UV maxima absorption of **6** at 260 nm indicated the presence of a double bond (C=C) conjugated with the carbonyl group. The IR spectrum revealed the presence of a COOH group (3414 cm^−1^), COOH (1687 cm^−1^), and C=C (1609 and 1633 cm^−1^). The presence of a conjugated carbonyl group was revealed by IR absorption, along with a resonance signal in the ^13^C-NMR spectrum at δ_C_ = 170.6.

The ^1^H-NMR spectrum of **6** showed two gem-methyl groups at δ_H_ 0.88 and 0.87 (each 3H, s, Me-12 and 13), one vinyl methyl group at δ_H_ 2.38 (3H, s, Me-15), two set methylene protons at δ_H_ 0.94 (1H, q-like, *J* = 12.0 Hz, H _ax_-4)/2.03 (1H, dtd, *J* = 12.0, 4.4, 2.0 Hz, H _eq_-4) and δ_H_ 1.14 (1H, t, *J* = 12.0 Hz, H_ax_-2)/1.75 (1H, ddd, *J* = 12.0, 4.4, 2.0 Hz, H_eq_-2), six methine moieties including one oxymethine proton at δ_H_ 3.81 (1H, tt, *J* = 12.0, 4.4 Hz, H-3), two mutually coupling methines at δ_H_ 1.61 (1H, m, H-5)/1.45 (t, *J* = 10.0 Hz, H-6), one olefinic proton at δ_H_ 5.73 (1H, br. s, H-10), and a trans pair of olefinic protons at δ_H_ 5.85 (1H, dd, *J* = 15.6, 10.0 Hz, H-7) and 6.08 (1H, d, *J* = 15.6 Hz, H-8), indicating that **6** was probably an apocarotenoid sesquiterpene [30] possessing a conjugated carbonyl group. The ^13^C-NMR and DEPT spectrum showed that **6** had a total of 15 carbons, with the skeleton consisting of 15 carbons, consistent with a sesquiterpenoid. The carbon of the sesquiterpene was assigned, from ^13^C-NMR and DEPT, as four methyls at δ_C_ 14.6 (C-15), 21.6 (C-14), 31.6 (C-12), and 22.0 (C-13), two methylenes at δ_C_ 44.8 (C-4) and 50.4 (C-2), one oxymethine carbon at δ_C_ 66.9 (C-3), two methines at δ_C_ 31.5 (C-5) and 58.2 (C-6), one olefinic carbon at δ_C_ 116.6 (C-10), two trans olefinic carbons at δ_C_ 135.6 (C-8) and 138.5 (C-7), and three quaternary carbons at δ_C_ 36.0 (C-1), 154.0 (C-6), and 170.6 (C-11). The above data also pointed to an apocarotenoid sesquiterpene skeleton.

The HMBC plot (Figure 2) showed the ^1^H,^13^C-NMR long-range correlations between the H atoms at δ_H_ 0.89 (Me-13) and the C atoms at δ_C_ 50.4 (C-2), 58.2 (C-6), and 31.6 (C-12), between the H atoms at δ_H_ 0.86 (Me-12) and the C atoms at δ_C_ 58.2 (C-6) and 22.0 (C-6), between the H atoms at δ_H_ 1.14/1.75 (CH_2_-2) and the C atoms at δ_C_ 31.6 (C-12) and 22.0 (C-13), between the H atoms at δ_H_ 0.94/2.03 (CH_2_-4) and the C atoms at δ_C_ 50.4 (C-2) and 21.6 (C-14), between the H atoms at δ_H_ 3.81 (H-3) and the C atoms at δ_C_ 50.4 (C-2) and 44.8 (C-4), between the H atom at δ_H_ 0.81 (H-14) and the C atoms at δ_H_ 44.8 (C-4), and 58.2 (C-6), between the H atom at δ_H_ 5.85 (H-7) and the C atoms at δ_H_ 58.2 (C-6) and 36.0 (C-1), between the H atom at δ_H_ 6.08 (H-8) and the C atoms at δ_H_ 116.6 (C-10) and 14.6 (C-15), and between the H atom at δ_H_ 2.28 (H-15) and the C atoms at δ_C_ 135.6 (C-8) and 116.6 (C-10). Furthermore, the ^1^H,^1^H-COSY (Figure 2) and ^1^H,^1^H-TOCSY spectra of **6** showed correlations for CH_2_-2/H-3, H-3/CH_2_-4, and CH_2_-4/CH-5, affording the fragment –CH_2_–CH(OH)–CH_2_–CH(Me)–CH–CH=CH–.

The relative stereochemistry of the three chiral centers (C-3, 5, and 6) of **6** was deduced from the splitting patterns and coupling constants of H-3 and from a nuclear Overhauser enhancement spectroscopy (NOESY) experiment (Figure 3). The results showed that H-3 (δ_H_ 3.81) was within an NOE distance from one methine (δ_H_ 1.61, H-5) and methyl (δ_H_ 0.86, H-12). The coupling constants of H-3 (*J* = 12.0, 4.4 Hz) coupled with the NOE experiment suggested that the H-3 and 5 were located at the axial position. The α-orientation of the methine group attached at δ_H_ 1.45 (H-6) was further supported by NOESY experiments (Figure 3), which showed the 1,3-diaxial interactions between CH-6 and both H_ax_-2 (δ_H_ 1.14) and H_ax_-4 (δ_H_ 0.94). The β-orientation of the C-6 (3-methylpenta-2,4-dienoic acid group) was evidenced by NOESY experiments, which also showed the correlations between H-8 (δ_H_ 6.08) and both H_ax_-6 (δ_H_ 1.45) and H-10 (δ_H_ 5.73), confirming the α-orientation of the proton on C-6. The E configuration of the 7,10-double bond was determined by the NOEs of Me-15 (δ_H_ 2.28) with H-7 (δ_H_ 5.85) and the NOEs of H-8 (δ_H_ 6.08) with both H-10 (δ_H_ 5.73) and H-6 (δ_H_ 1.45). Other significant NOE correlations were also observed between Me-13 and CH_2_-2, H-6, and Me-12, between H_eq_-2 and Me-12 and 13, and between H_eq_-4 and H_ax_-3 (Figure 3). From these data, the structure of compound **6** was thus deduced to be rel-(2E,4E)-5-((3S,5R,6R,)-4-hydroxy-2,2,6-trimethylcyclohexyl)-3-methylpenta-2,4-dienoic acid and named cimicitaiwanin F.

In summary, the structure obtained above belongs to the 9,10-seco-cimigenol-type triterpenoid with an ether bond between C-3 and C-10, which is different from the cyclolanostane-type triterpenoids in the previous literature [16,17,18,19,20]. C. taiwanensis is endemic to Taiwan, and it is worth continuing to develop this type of skeleton in the future. This is the first time one monocyclofarnesane-type sesquiterpene of cimicitaiwanin F has been isolated from Ranunculaceae plants. These findings provide meaningful chemotaxonomic information.

### 2.2. Biological Studies

The anti-inflammatory activities of the 5 isolates (**2**–**6**) in sufficient amounts were evaluated by examining their inhibitory effects on lipopolysaccharide (LPS)-induced inducible nitric oxide synthase (iNOS)-dependent NO 6 production in the murine macrophage cell line RAW 264.7 (Table 4). From the results of our anti-inflammatory tests, the following conclusions could be drawn.

Compared with quercetin (IC_50_ value 34.58 μM), which was used as a positive control in this study, cimicitaiwanins B–F (**2**–**6**, resp.) exhibited NO inhibitory activity with *IC*_50_ values of 8.37, 6.54, 10.11, 24.58, and 15.36 μM, respectively. Compounds **2**–**4** and **6** showed about four-, five-, three-, and twofold NO inhibitory activities compared with quercetin, respectively, while **5** showed moderate NO inhibitory activity. Among the 9,19-cyclolanostane triterpene analogs, compound **3** (cimicitaiwanin C) with C=O at the C-11 and OH-7 groups exhibited more effective inhibition than its analogue compound **2** (cimicitaiwanin B, with the CH_2_–7 and CH_2_–11 groups) and compound **4** (cimicitaiwanin D, with a xyloside at C-3) against LPS-induced NO generation. Compound **5** (cimicitaiwanin E), with C=O at C-3 conjugated with a double bond at C-1/2, exhibited less effective inhibition than similar analogues. The cytotoxic effects were tested using an MTT experiment. The high cell viability (95, 91, 89, and 87%) of compounds **3**–**6** at 50 μM, respectively, showed that their inhibitory activities against LPS-induced NO generation did not arise from their cytotoxicities. In contrast, compound **2** (cimicitaiwanin B) with an IC_50_ value of 8.37 μM also showed inhibition of NO production of macrophages, but the low cell viability (<80%) suggested the possibility of cytotoxicity. Based on the analyses of these data, we may propose that for 9,10-seco-cimigenol-type triterpene, hydrophobic groups, such as carbonyl and acetoxy, instead of a hydroxyl group at C-3, C-11, or C-25, are essential for anti-inflammatory activity.

After comparison with the plants of the same genus and having anti-inflammatory potential in the past [31,32,33], it is interesting to find that the substituents at these key positions (C-3, C-11, or C-25) lost their anti-inflammatory activity when combined with other similar cycloartane skeletons. In short, other anti-inflammatory active substances of the same genus need to be substituted by a hydrophilic group OH or a xylose group. The existence of an acetoxy group at C-3, C-11, or C-25 and the absence of an oxygen-bearing group at C-15 are essential for the NO production inhibitory activity of cycloartane triterpenoids and nortriterpenoids in LPS-stimulated RAW264.7 cells.

The same genus has also been reported to have anti-inflammatory active metabolites in the past, such as five undescribed cycloartane triterpenoids and nortriterpenoids from *Actaea vaginata* with anti-inflammatory effects in LPS-stimulated RAW264.7 macrophages [31] and 12 cycloartane compounds isolated from *Cimicifuga dahurica* (Turcz.) Maxim., which is traditionally used as an antipyretic and analgesic in Korea, which were found to inhibit the production of pro-inflammatory cytokines (IL-12p40, IL-6, and TNF-α) in vitro in bone marrow-derived dendritic cells stimulated with LPS [32], as well as five cycloart-7-ene triterpenoid glycosides isolated from the roots of *Cimicifuga dahurica*, which can reduce the release of NO in a dose-dependent manner in LPS-stimulated RAW 264.7 macrophages [33].

Most of the compound skeletons isolated this study belong to 9,10-seco-cimigenol-type triterpene with an ether bond between C-3 and C-10, between C-16 and 23, and between C-16 and 24 on the E/F ring, which is different from the cimicifugenol type derivatives of the same genus in the literature that also have anti-inflammatory activity.

## 3. Materials and Methods

### 3.1. General Experimental Procedures

The melting points were determined on a Yanaco micro-melting point apparatus (Yanaco, Tokyo, Japan) and were uncorrected. The general method was to the heat the sample indirectly by placing the prepared sample (either packed in a glass capillary or on a glass cover slip) in or on a heated medium. These days, this is most commonly a heated metal block such as a Mel-Temp apparatus. There are other designs, such as the Fisher-Johns apparatus. A more basic but just as effective method is the Thiele tube method, where the capillary is immersed in a heated oil bath. Note that the Thiele tube system is also used for boiling point determination.

Optical rotations were measured using a Jasco DIP-370 polarimeter (Jasco, Easton, MD, USA) in CHCl3. 1. We checked that the lamp was turned on before analysis (looking for a spot of light on the front left side). If not, we (1) opened the Spectra Manager acquisition program through the desktop, (2) clicked on the Environment icon on the right side under the Measurement section, which opened an Environment Settings window, and (3) selected the Light Source tab and checkmark to turn the internal source ON. Then, we checked the wavelength of the filter in use by lifting the circular plate on the top lower left-hand side, where 589 nm was the standard on the sodium D line. We also had a 365-nm filter available and could ask the facility manager for assistance. Next, we set the beam width filter inside the sample compartment to 3 or 8, depending on the cell diameter. Typically, 10-mm cells are used so set the beam width to 8. In the fourth step, we opened the Spectra Manager acquisition program through the desktop. On the right side under the Measurement section, we clicked on the Standard icon. This opened an Optical Rotation window. Then, under the {Instrument} menu, we clicked on Start Analyzer. This should start measuring the optical rotation within the left-hand sidebar. Then we needed to zero the background through the {Measure} menu by selecting Zero Clear. The optical rotation measurement should read all zeroes. Next, we took measure of the blank or solvent, which involved (1) placing the cell with just the solvent into the cell compartment and using straps to secure the cell, (2) opening the Measurement Parameters window through the {Measure} menu and selecting Start, (3) setting the following parameters: Integration = 1 s, Repeat = 10 scans, Interval = 1 s, and Measurement Mode = optical rotation and then unchecking the temperature box, and (4) clicking OK. The instrument would take 10 scans and output the average and statistical information. This blank value should be close to zero. This step could be repeated with another sample. The optical rotation measurement limits were between 0.0002° and ±90°. If the reading disappeared when inserting the sample, this indicated that the sample was absorbing too much light, and the concentration should be reduced before trying again. When finished, in the {Instrument} menu we clicked Stop Analyzer, followed by saving or printing any data desired. Finally, when completely done, we exited the program in the following order: (1) closing the Optical Reaction window through {Measure} → Exit, (2) closing the Spectra Manager window through {Application} → Exit, and (3) removing the sample from the chamber and turning off the Polarimeter P1010.

Ultraviolet (UV) spectra were recorded with a Jasco UV-240 spectrophotometer (Jasco, Easton, MD, USA) from 200 to 800 nm with a scanning rate of 400 nm min^−1^ at 25 °C. The spectra of the cell sap were measured in a quartz cell with a path length of 0.1 mm, and those of the reproduced solutions were tested in a quartz cell with a 10-mm path length.

Infrared (IR) spectra (neat or KBr) were recorded on a Perkin Elmer 2000 FT-IR spectrometer (PerkinElmer, Waltham, MA, USA). For equipment safety purposes, the instrument was constantly switched on and purged. Please, when using this device, do not change the flow of purging. Instead, (1) double click on Spectrum software. (2) Make sure that the sample compartment is empty. If opened, let the sample compartment be purged for at least 2 min. (3) In the Instrument menu, select Scan Background and check the parameters (single beam, background filename, start and end scan, resolution, etc.). Click OK and overwrite the old background. (4) Insert the sample into the sample compartment and wait for 2 min. (5) In the Instrument menu, select Scan Sample, then check the parameters (ratio, filename, start and end scan, resolution, etc.) and click OK. (6) To find the peaks, click the Label Peaks button. To adjust the threshold, right-click Label Peaks again to enter new values. (7) To expand the region, left-click and drag to select, and then double click. (8) To save the file, select Save As in the File menu. (9) When finished, close the Spectrum software and logout. (10) Do not turn off the FTIR. Do not turn off the computer.

Nuclear magnetic resonance (NMR) spectra, including correlation spectroscopy (COSY), nuclear Overhauser effect spectrometry (NOESY), rotating frame nuclear Overhauser effect spectrometry (ROESY), heteronuclear multiple-bond correlation (HMBC), and heteronuclear single-quantum coherence (HSQC) experiments, were acquired using a Varian Mercury-500 and Varian-Unity-Plus-400 instrument and reported in CDCl_3_. ^1^H and ^13^CNMR chemical shifts were reported in ppm relative to either TMS (^1^H) (δ = 0 ppm, *J* in Hz) as an internal standard or the residual solvent peak as follows: CDCl_3_ = 7.26 (^1^H NMR) and CDCl_3_ = 77.0 (^13^C NMR), with type nt = 8, 16, or 32, according to sample concentration (or use a multiple of 8 for nt) and type ga (submit to acquisition) to begin acquisition.

Electrospray ionization (ESI) and high-resolution electrospray ionization (HRESI) mass spectra were recorded on a Bruker APEX II mass spectrometer (Bruker, Bremen, Germany). ESI is a technique for generating ions for mass spectrometry using electrospray by applying a high voltage to a liquid to produce an aerosol. Due to relatively fragile biomacromolecules, their structures are easily destroyed during the process of dissociation and ionization. ESI overcomes the tendency of these molecules to fragment upon ionization. ESI differs from other atmospheric pressure ionization processes in that it may produce multiple charge ions, effectively extending the analyzer’s mass range to accommodate the magnitude of kDa-mDa observed by proteins and their associated peptides.

Silica gel (70–230, 230–400 mesh) (Merck, Darmstadt, Germany) was used for column chromatography (CC). Silica gel 60 F-254 (Merck, Darmstadt, Germany) was used for thin-layer chromatography (TLC) and preparative thin-layer chromatography (PTLC), and spots were visualized by heating after spraying with 10% aqueous H_2_SO_4_. Spherical C18 100A reversed-phase silica gel (RP-18) (particle size: 20–40 μm) (Silicycle) was used, as was an HPLC: Spherical C18 column (250 × 10 mm, 5μm) (Waters) and LDC-Analytical-III apparatus. Acetonitrile and water at a 10:1 was the mobile phase, with a flow rate of 1.0 mL/min.

### 3.2. Plant Material

The roots of *Cimicifuga taiwanensis* were collected from Huagang Mountain in Nantou County, Taiwan in August 2015. The plant was identified by Dr. Shang-Tzen Chang, Professor of the Department of Forestry at National Taiwan University. A voucher specimen (Kuo-19988) was deposited in the herbarium of the Department of Botany of National Taiwan University in Taipei, Taiwan.

### 3.3. Isolation and Characterization of Secondary Metabolites

Air-dried roots of *Cimicifuga taiwanensis* (10 kg) were extracted five times with MeOH (50 L) at r.t. (7 days thrice). The methanol extract was concentrated, with the brown residue suspended in H_2_O (7 L) and then extracted with EtOAc, with the EtOAc fraction (405 g) subjected to CC (silica gel, hexane/EtOAc of increasing polarity, and EtOAc/MeOH gradient) to get 10 fractions (Fractions 1–10), with each product fraction further purified by HPLC. Fraction 3 was subjected to MPLC (RP-18; MeOH/H_2_O 1:2) to produce four fractions: fractions 3.1–3.4. Fraction 3.2 was subjected to HPLC (RP-18; E/H = 30%, E/D = 20%, M/D = 1%) to obtain cimicitaiwanin E (**5**, 7.0 mg). Fraction 3.4 (81.4 mg) was applied to RP-18 preparative TLC developed with MeOH–H_2_O (3:1) to afford **6** (2.7 mg). Cimicitaiwanins A‒D (**1**‒**4**; 2.1, 4.5, 2.9, and 8.5 mg) were obtained with 30% EtOAc/hexane (CC) from fraction 5 (14.9 g) and purified by HPLC (10% EtOAc/CH_2_Cl_2_).

Cimicitaiwanin A (**1**): needles; mp. 228–230 °C; 
$\alpha_{D}^{26}$
 = +9.52 (*c* 0.21, CHCl_3_); IR (Neat): 3452 (-OH) cm^−1^; ^1^H NMR (500 MHz, CDCl_3_) (see Table 1); ^13^C NMR (125 MHz, CDCl_3_) (see Table 2); EIMS (70 eV) *m*/*z* (%): 502 ([M]^+^, 100), 484 (55), 469 (36), 451 (24); HREIMS *m*/*z* 502.3303 [M]^+^ (calculated for C_30_H_46_O_6_, 502.3289).

Cimicitaiwanin B (**2**): needles; mp. 177–179 °C; 
$\alpha_{D}^{26}$
 = +32.44 (*c* 0.45, CHCl_3_); IR (Neat): 3452 (-OH), 1737 (ester C=O) cm^−1^; ^1^H NMR (500 MHz, CDCl_3_) (see Table 1); ^13^C NMR (125 MHz, CDCl_3_) (see Table 2); EIMS (70 eV) *m*/*z* (%): 544 ([M]^+^, 100), 526 (39), 484 (35), 466 (31), 451 (30); HREIMS *m*/*z* 544.3377 [M]^+^ (calculated for C_32_H_48_O_7_, 544.3395).

Cimicitaiwanin C (**3**): needles; mp. 104–105 °C; 
$\alpha_{D}^{26}$
 = +10.07 (*c* 0.29, CHCl_3_); UV (MeOH): 255 (3.85) nm; IR (Neat): 3419 (OH), 1732 (ester C=O), 1661 (conjugated C=C) cm^−1^; ^1^H NMR (500 MHz, CDCl_3_) (see Table 1); ^13^C NMR (125 MHz, CDCl_3_) (see Table 2); EIMS (70 eV) *m*/*z* (%):588 ([M]^+^, 14), 542 (14), 498 (100), 483 (28), 480 (19), 465 (18); HREIMS *m*/*z* 588.3185 [M]^+^ (calculated for C_32_H_46_O_8_, 588.3187).

Cimicitaiwanin D (**4**): yellow-green oily. 
$\alpha_{D}^{26}$
 = +9.52 (*c* 0.21, CHCl_3_); IR (Neat): 3426 (OH), 1731 (ester C=O), 1665 (double bond) cm^−1^; ^1^H NMR (500 MHz, CDCl_3_) (see Table 1); ^13^C NMR (125 MHz, CDCl_3_) (see Table 2); EIMS (70 eV) *m*/*z* (%): 660 ([M]^+^, 1), 528 (28), 510 (39), 468 (58), 450 (90), 435 (85), 297 (38), 170 (100), 105 (75); HREIMS *m*/*z* 660.3836 [M]^+^ (calculated for C_37_H_56_O_10_, 660.3868).

Cimicitaiwanin E (**5**): yellow needles; mp. 224–226 °C; 
$\alpha_{D}^{26}$
 = −54.22 (*c* 0.7, CHCl_3_); UV (MeOH): 250 (3.84), 290 (3.59)(sh) nm; IR (Neat): 3508 (OH), 1668, 1610 (conjugated C=O) cm^−1^; ^1^H NMR (500 MHz, CDCl_3_) (see Table 1); ^13^C NMR (125 MHz, CDCl_3_) (see Table 2); EIMS (70 eV) *m*/*z* (%): 482 ([M]^+^, 100), 467 (36), 464 (41), 449 (59), 322 (28); HREIMS *m*/*z* 482.3027 [M]^+^ (calculated for C_30_H_42_O_5_, 482.3027).

Cimicitaiwanin F (**6**): oil; 
$\alpha_{D}^{26}$
 = −9.70 (*c* 0.27, CHCl_3_); UV (MeOH): 260 (3.98) nm; IR (Neat): 3414 (OH), 1687 (conjugated C=O) cm^−1^; ^1^H NMR (500 MHz, CDCl_3_) (see Table 1); ^13^C NMR (125 MHz, CDCl_3_) (see Table 2); EIMS (70 eV) *m*/*z* (%): 252 ([M]^+^, 8), 234 (13), 178 (12), 152 (48), 111 (24), 107 (100); HREIMS *m*/*z* 252.1727 [M]^+^ (calculated for C_15_H_24_O_3_, 252.1720). 

### 3.4. Determination of NO Production and Cell Viability Assay

The mouse macrophage cell line (RAW 264.7) was obtained from the Bioresource Collection and Research Center (BCRC 60001) and cultured at 37 °C in Dulbecco’s Modified Eagle’s Medium (DMEM) supplemented with 10% fetal bovine serum (Gibco), 4.5 g/L glucose, 4 mM glutamine, penicillin (100 units/mL), and streptomycin (100 μg/mL) in a humidified atmosphere in a 5% CO_2_ incubator. The cells were treated with 10, 25, and 50 μM of natural products in the presence of 1 μg/mL lipopolysaccharide (LPS, Sigma-Aldrich) for 20 h. The concentration of NO in the culture supernatants was determined to be nitrite, a major stable product of NO, by a Griess reagent assay [34], and cell viabilities were determined using the MTT assay as described previously [35].

### 3.5. Statistical Analysis

Results are expressed as the mean ± SEM, and comparisons were made using Tukey’s HSD test. A probability of 0.05 or less was considered significant. SigmaPlot software was used for the statistical analysis.

## 4. Conclusions

Plants are an important source for the discovery of new products with medicinal value for drug development, and plant secondary metabolites are a unique source of pharmaceutical food additives, flavor, and other industrial values [36,37]. Cimicifuga is the dry rhizome of the perennial herb of Ranunculaceae, which is mainly produced in Liaoning, Heilongjiang, Hunan, and Shanxi in China. Its functions are to resolve the exterior, vent rashes, clear heat, resolve toxins, and raise yang qi. (Yin Qi vs Yang Qi. Most Chinese medicine practitioners think that there are many kinds of qi (qì), and the most basic kinds are yin qi and yang qi. Everything is a balance of yin and yang. Yin is female, dark, and formless. Yang is male, light, and has a form. Females have more yin qi, while males have more yang qi, and as people age, they lose qi.)

This can be applied to wind heat headaches, toothaches, mouth sores, throat swelling pain, measles with no outthrust, spots due to the yang toxin (the same meaning as yang poisoning, as the disease name refers to common symptoms such as redness and spots on the face due to the infection of the virus), rectal prolapse and prolapse of the uterus in Chinese folk medicine and is included in the Chinese Pharmacopoeia. *Cimicifuga taiwanensis* is endemic to Taiwan, and as there is no report on its chemical and biological activity in the past, we chose it for research on active constituents. In summary, we isolated and characterized six undescribed derivatives—cimicitaiwanins A–F—from the methanol extract from the roots of *C. taiwanensis* that was collected from Huagang in Nantou County in the middle of Taiwan. The relative configurations of the new isolates were determined by comparing their optical activities with related derivatives and NOESY plots. Cimicitaiwanins C, D, E, and F showed inhibitory activities against LPS-induced NO production in RAW 264.7. Most of the compounds discovered in this research were 9,10-seco-cimigenol-type triterpene with an ether bond between C-3 and C-10. The potential action mechanism of the potent cimicitaiwanins C, D, E, and F (**3**‒**6**) need to be studied further on pro-inflammatory cytokines, chemokines, and inducible enzymes by immunostaining, ELISA, and western blotting methods. It is interesting to note that the apocarotenoid isolated in this study was monocyclofarnesane-type sesquiterpene, and they had never before been isolated from other Formosan Ranunculaceae plants. These results provide meaningful chemotaxonomic information.

Cimicifugae rhizome has been used in cooling and detoxification and as an antipyretic and analgesic agent for the treatment of some types of headaches and toothaches in Chinese folk medicine, as well as being included in the Chinese Pharmacopoeia. In the past, *Cimicifuga* spp. have focused on anti-cancer constituent research [37,38,39,40,41], and there are few reports of anti-inflammatory active substances. The active structure obtained this time belonged to 9,10-seco-cimigenol-type triterpene with an ether bond between C-3 and C-10, which is different from the past cyclolanostane-type triterpenoid compositions in the literature. *C. taiwanensis* is endemic to Taiwan, and it is worth continuing to develop this type of skeleton in the future.

## Figures and Tables

**Figure 1 molecules-27-01657-f001:**
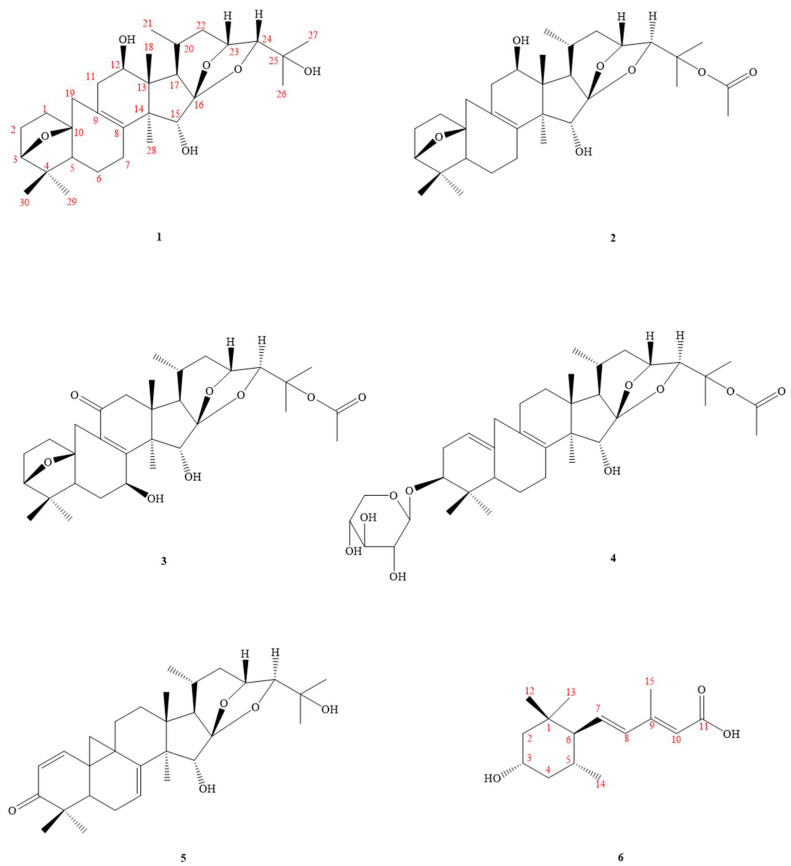
Compounds **1**–**6**, isolated from *Cimicifuga taiwanensis*.

**Figure 2 molecules-27-01657-f002:**
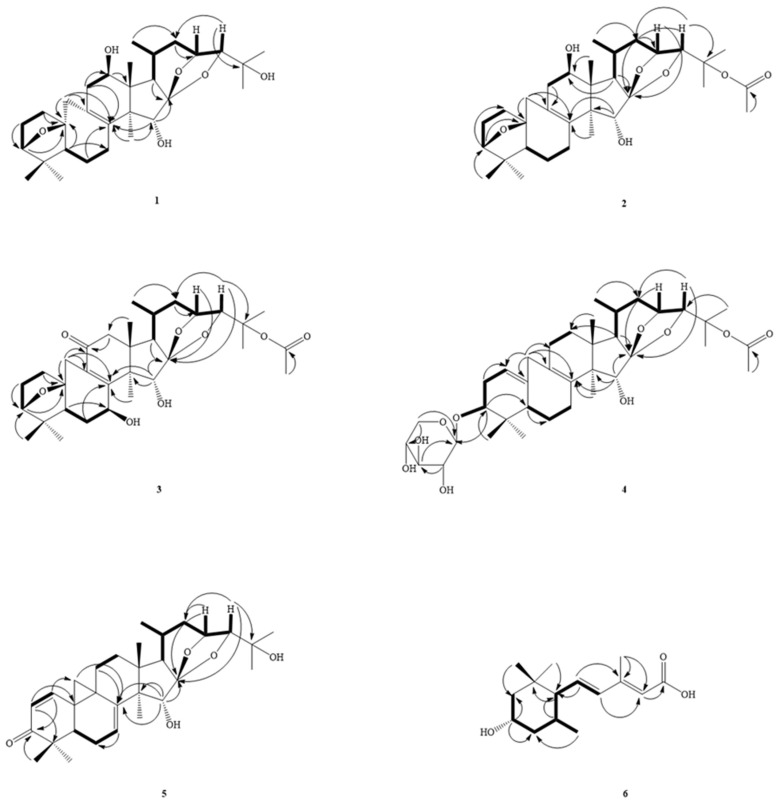
Key COSY (▬) and HMBC (→) correlations of **1**‒**6**.

**Figure 3 molecules-27-01657-f003:**
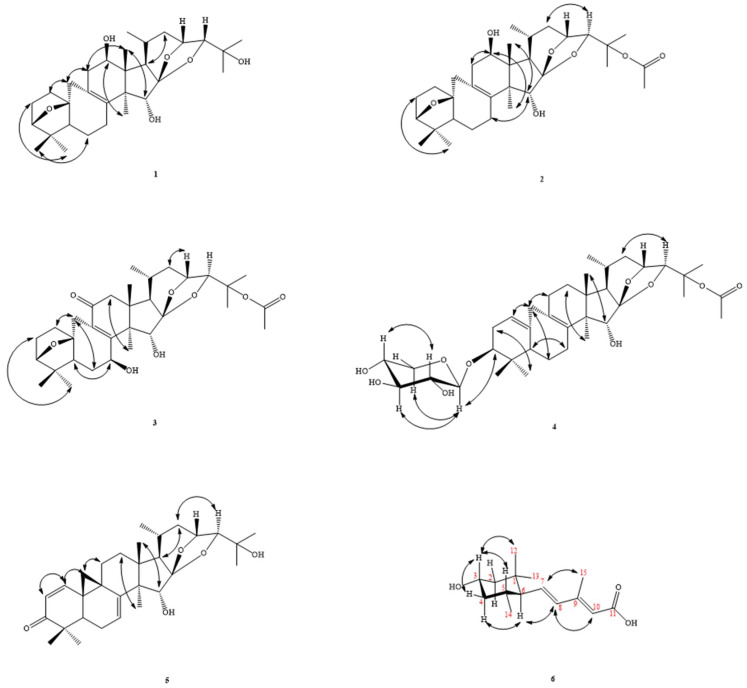
Major NOESY (↔) contacts of **1**‒**6**.

**Table 1 molecules-27-01657-t001:** ^1^H NMR data for compounds **1**–**6** in CDCl_3_ (*δ* in ppm and *J* in Hz at 500 MHz in CDCl_3_).

No	1	2	3	4	5	6
1	1.40 (m, H-1β) 1.58 (m, H-1α)	1.39 (m) 1.57 (m)	1.52 (td, *J* = 12.0, 4.8, H-1β), 1.18 (m, H-1α)	5.20 (br s)	6.68 (d, *J* = 9.8)	
2	1.71 (m, H-2β) 1.93 (m, H-2α)	1.68 (m, H-2β) 1.91 (m, H-2α)	1.70 (m, H-2β) 1.88 (m, H-2α)	2.10 (m, H-2β) 2.33 (m, H-2α)	5.99 (d, *J* = 9.8)	1.14 (t, *J* = 12.0, H_ax_-1α), 1.75 (ddd, *J* = 12.0, 4.4, 2.0, H_eq_-1β)
3	3.72 (d, *J* = 5.6)	3.71 (d, *J* = 5.6)	3.80 (d, *J* = 5.2)	3.41 (dd, *J* = 8.6, 5.4)		3.81 (tt, *J* = 12.0, 4.4, H_ax_-3β)
4						0.94 (q-like, *J* = 12.0, H_ax_-4α), 2.03 (dtd, *J* = 12.0, 4.4, 2.0, H_eq_-4β)
5	1.15 (m)	1.15 (d, *J* = 12.0)	0.93 (m)	1.95 (m)	1.92 (m)	1.61 (m)
6	1.68 (m, H-6β) 1.52 (m, H-6α)	1.48	1.84 (m, H-6β) 2.00 (m, H-6α)	1.46 (m, H-6β) 1.78 (m, H-6α)	1.76 (m, H-6β) 2.03 (m, H-6α)	1.45 (t, *J* = 10.0)
7	2.42 (m)	2.42 (m)	4.79 (dd, *J* = 10.2, 7.6)	2.58 (m, H-7β) 2.43 (m, H-7α)	5.74 (dd, *J* = 7.4, 1.8)	5.85 (dd, *J* = 15.6, 10.0)
8						6.08 (d, *J* = 15.6)
9						
10						5.73 (br s)
11	1.84 (m, H-11β) 2.59 (m, H-11α)	1.82 (m, H-11β) 2.59 (m, H-11α)		2.03 (m, H-11β) 1.24 (m, H-11α)	1.44 (m, H-11β) 2.30 (m, H-11α)	
12	4.10 (t, *J* = 7.6, H-12α)	4.19 (m)	2.43/2.72 (each d, *J* = 17.6, H-12β/α)	1.48 (m, H-12β) 1.84 (m, H-12α)	1.78 (m, H-12β) 1.86 (m, H-12α)	0.86 (s)
13						0.87 (s)
14						0.81 (d, *J* = 6.8)
15	4.02 (d, *J* = 7.6, H-15β) 2.77 (d, *J* = 7.6, OH)	4.10 (d, *J* = 7.6)	4.74 (br s)	4.07 (s)	4.06 (s)	2.28 (d, *J* = 1.2)
16						
17	1.47 (d, *J* = 1.4)	1.54 (m)	1.56 (d, *J* = 11.2)	1.34 (m)	1.33 (d, *J* = 10.9)	
18	0.83 (s)	0.82 (s)	1.05 (s)	0.83 (s)	1.03 (s)	
19	1.69 (br d, *J* = 13.6), 3.14 (br d, *J* = 13.6)	1.67 (br d, *J* = 13.6), 3.12 (br d, *J* = 13.6)	2.69/3.15 (each d, *J* = 13.2),	2.41 (br d, *J* = 14.8), 2.83 (br d, *J* = 14.8)	1.11/1.66 (each d, *J* = 4.0)	
20	1.51 (m)	1.52 (m)	1.68 (m)	1.64 (m)	1.61 (m)	
21	1.08 (d, *J* = 6.4)	1.08 (d, *J* = 5.2)	0.91 (d, *J* = 6.8)	0.91-(d, *J* = 6.8)	0.90 (d, *J* = 6.8)	
22	2.00 (m, H-22β) 2.15 (m, H-22α)	2.35 (m, H-22β) 1.01 (m, H-22α)	2.35 (m, H-22β) 1.01 (m, H-22α)	2.31 (m, H-22β) 0.97 (m, H-22α)	1.96 (m, H-22β) 2.16 (m, H-22α)	
23	4.42 (ddd, *J* = 9.6, 4.0, 3.2)	4.37 (d, *J* = 9.2)	4.40 (br d, *J* = 9.2)	4.37 (d, *J* = 8.4)	4.45 (ddd, *J* = 9.8, 4.4, 2.6)	
24	3.54 (d, *J* = 4.0)	3.90 (s)	3.93 (s)	3.90 (s)	3.57 (s)	
25						
26	1.22 (s)	1.41 (s)	1.41 (s)	1.41 (s)	1.21 (s)	
27	1.32 (s)	1.45 (s)	1.46 (s)	1.45 (s)	1.32 (s)	
28	0.92 (s)	0.93 (s)	1.15 (s)	0.94 (s)	1.07 (s)	
29	0.99 (s)	0.97 (s)	0.99 (s)	0.99 (s)	1.13 (s)	
30	0.94 (s)	0.91 (s)	0.98 (s)	0.78 (s)	1.02 (s)	
OAc		1.97 (s)	1.99 (s)	1.98 (s)		
OAc						
1′				4.43, (d, *J* = 5.9)		
2′				3.45, (dd, *J* = 7.2, 5.9)		
3′				3.58 (t, *J* = 7.2)		
4′				3.72 (td, *J* = 7.2, 4.4)		
5′				4.02 (dd, *J* = 12.2, 4.4, H-5′β) 3.33 (m, *J* = 12.2, 7.2, H-5′α)		

**Table 2 molecules-27-01657-t002:** ^13^C NMR data for compounds **1**–**5** (*δ* in ppm at 125 MHz for ^13^C NMR in CDCl_3_).

No	1	2	3	4	5	6
1	36.2	36.4	35.3	116.9	151.6	36.0
2	25.4	25.8	25.4	31.1	126.9	50.4
3	84.9	84.8	86.0	85.1	203.9	66.9
4	45.2	45.4	44.9	38.1	45.2	44.8
5	54.6	54.7	50.3	50.0	40.1	31.5
6	22.5	22.9	32.9	25.2	21.9	58.2
7	30.2	30.5	69.7	29.0	115.3	138.5
8	136.4	136.1	159.0	136.3	145.7	135.6
9	124.6	124.1	130.1	126.8	26.2	154.0
10	89.6	89.5	88.3	137.8	32.8	116.6
11	43.4	43.7	198.5	29.4	26.7	170.6
12	71.0	71.1	49.2	31.9	33.6	31.6
13	46.0	46.3	42.2	41.4	41.1	22.0
14	52.2	52.2	49.9	49.4	50.9	21.6
15	74.8	74.5	73.9	74.8	77.5	14.6
16	112.2	111.9	111.0	112.0	111.5	
17	59.3	57.2	57.1	57.0	61.1	
18	9.7	10.3	18.8	17.6	21.4	
19	35.0	35.2	25.6	41.6	30.6	
20	22.7	23.6	23.7	24.2	23.2	
21	20.4	21.6	19.4	20.4	19.6	
22	29.5	38.0	37.3	38.0	29.3	
23	73.3	71.8	72.0	72.0	73.6	
24	82.3	86.0	86.8	86.1	83.2	
25	68.7	82.3	82.5	82.5	68.8	
26	24.1	22.4	22.4	22.1	24.4	
27	31.3	23.3	22.9	23.4	31.7	
28	16.2	16.6	18.1	17.9	18.1	
29	25.1	25.4	25.3	25.6	21.7	
30	23.3	23.6	23.5	16.8	19.2	
OAc		22.8	22.5	22.8		
OAc		169.6	170.4	169.6		
1′				104.1		
2′				72.5		
3′				74.1		
4′				69.7		
5′				63.9		

**Table 3 molecules-27-01657-t003:** 2D-NMR data for compounds **1**–**6** in CDCl_3_ (*δ* in ppm, *J* in Hz at 500 MHz in CDCl_3_).

No	1			2			3			4			5			6		
	COSY	HMBC	NOESY	COSY	HMBC	NOESY	COSY	HMBC	NOESY	COSY	HMBC	NOESY	COSY	HMBC	NOESY	COSY	HMBC	NOESY
1		3, 5, 19	2β, 19	2	3, 19		2	3, 19			2, 19	2, 19	2	19	2, 11		2, 6, 12, 13	
2	1	1, 3		1, 3	5		1, 3			1	3	1′, 3	1		1	3	12, 13	12, 13
3	2	2, 29, 30	2, 29, 30	2	29, 30	2β	2	29, 30			1′	1′, 2α, 5,		1, 29, 30		2, 4	2, 4	12
4		2, 5, 18, 28			5, 29, 30			29, 30			2, 29, 30			2, 5, 29, 30		3	2, 14	
5		1, 3, 19, 29, 30	6α		1, 3, 19, 29			3, 6, 19, 29, 30	1α		3, 19, 29, 30	7α, 29		1, 6, 7, 19, 29, 30		6	4, 6	12
6	7	5		7				5, 7		7	5			7	19, 29	5, 7	2, 7, 12, 13	2β, 4β
7	6	5		6	5, 19		6	5, 6	5, 6α, 28	6				6	6α, 15	6, 8	6	15
8		6, 11, 15, 19, 28			19, 28			6, 7, 15, 19, 28			15, 19, 28			6, 11, 15, 19, 28		7	10, 15	6, 10
9		11, 19			11, 19			17, 19			11, 12, 19			1, 7, 11, 12, 19			7, 15	
10		1, 3, 5, 19			3, 5, 19			1, 3, 6, 19			1, 5, 6, 19			2, 5, 6, 19		15	15	8
11		11		12	19			12, 19		12	12, 19	12β	12	12	12, 19 28		10, 15	
12	11	11, 17, 18	11α, 28	11	17, 18	11α, 17, 28		17, 18		11	18, 20	11		17, 18			6, 13	
13		12, 17, 18, 28			12, 17, 18, 20			12, 17, 18, 28			12, 17, 18, 20, 28			11, 12, 17, 18, 28			2, 6, 12	
14		18, 28			18, 28			12, 15, 18, 28			12, 15, 18, 28			7, 12, 15, 18, 28		5	4, 6	
15		28			28	7β, 18		28			28			28	7, 28	10	8, 10	7
16		15, 17			12, 17, 23, 24			15, 17, 23, 24			15, 17, 23, 24			15, 17				
17		12, 18, 21, 22	22α		12, 18, 21, 22			18, 21, 22	12α, 22α	20	18, 22, 28			21, 22	12α, 20, 22α, 28			
18		12, 17	11β, 15		12, 17	15β			12β		17	11β, 12β, 15, 20		12	12β, 20			
19		1	11β						6β, 1β			1, 11β, 6β		11	11β			
20	21	17, 21	22		17, 21, 22, 23			17, 21		17, 21, 22	22	22α		21, 22				
21		17	22							20		22α		20, 22	17, 20, 22α			
22		20, 21, 24			21, 24			20, 21, 24		23	20, 21, 24	24 (22α) 20 (22β)		20, 21, 24	20, 21, 17, 26			
23	22β, 24	22, 24	22	22	20, 22	22β	22	22	22β	22	22			22, 24	22, 24, 26			
24	23	22, 26, 27			22, 26, 27	22α		22, 26, 27			22, 26, 27	17, 22α		22, 26, 27	23, 26, 27			
25		24, 26, 27			24, 26, 27			26, 27			24, 26, 27			24, 26, 27				
26		27			24, 27						27			27				
27		24, 26			26						26			26				
28		15	12			2α			12α		15	7α, 12α, 17		15	7, 11α, 12α			
29		3, 5, 30	2α		30	2α		30	2α		30	1′, 2α, 5, 6α,13		30				
30		3, 29	6β		29			3, 29			3, 29	6		5, 29	6β, 19			
OAc																		
OAc					OAc			OAc			OAc							
1′											2′, 3′, 5′, 3	3′, 5′α, 3						
2′											3′	4′						
3′											2′, 4′, 5′	1′						
4′											3′ 5′	2′						
5′											1′, 3′							

**Table 4 molecules-27-01657-t004:** Inhibitory effects of the 5 isolates (**2**–**6**) from *Cimicifuga taiwanensis* on LPS-activated NO production in RAW 264.7 macrophages.

Compounds	IC_50_ (μM) ^(a)^	
	NO	Cell Viability (% Control)
Cimicitaiwanin B (**2**)	8.37 ± 3.25 *	69.14 ± 2.87
Cimicitaiwanin C (**3**)	6.54 ± 0.87 *	95.53 ± 3.39
Cimicitaiwanin D (**4**)	10.11 ± 0.47 *	91.12 ± 4.22
Cimicitaiwanin E (**5**)	24.58 ± 4.97 **	88.54 ± 1.50
Cimicitaiwanin F (**6**)	15.36 ± 0.85 **	86.84 ± 2.95
Quercetin ^(b)^	34.58 ± 2.34 *	95.56 ± 1.91

^(a)^ The IC_50_ values were calculated from the slope of the dose–response curves (SigmaPlot). Values are expressed as mean ± SEM (*n* = 4) of 3 independent experiments. * *p* < 0.05. ** *p* < 0.01, compared with the control. ^(b)^ Quercetin was used as a positive control.

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
