# Peer review of "Secondary Metabolites with Anti-Inflammatory from the Roots of Cimicifuga taiwanensis"

_molecules, 2022, doi:10.3390/molecules27051657_

Round 1

Reviewer 1 Report

The aim of this paper was to investigate some secondary metabolites with anti-inflammatory activity from the roots of Cimicifuga taiwanensis. The manuscript fits within the scope of the journal. The manuscript is interesting.  The title is clear and it is adequate to the content of the article. The conclusions or summary are supported by the content.

Some revisions are necessary to improve the clarity of the presentation:

- Please include in the abstract the aim of your study and the background.

- Please highlight the degree of novelty and originality of the work.

- Are the results obtained in triplicate? Include more statistical information. Please include a subchapter for statistical analysis!

I didn’t see the citations 6-10 in the text.

-The anti-inflammatory potential needs to be compared with other compounds. The scientific dialogue is quite weak.

- Include in the text potential research directions. What are the future applications? What are the next research directions?

Author Response

Reviewer 1

The aim of this paper was to investigate some secondary metabolites with anti-inflammatory activity from the roots of Cimicifuga taiwanensis. The manuscript fits within the scope of the journal. The manuscript is interesting. The title is clear and it is adequate to the content of the article. The conclusions or summary are supported by the content.

Some revisions are necessary to improve the clarity of the presentation:

- Please include in the abstract the aim of your study and the background.

Response- Thank you for your comments. The aim of our study and the background have included in the abstract. (Highlighted in yellow colors)

- Please highlight the degree of novelty and originality of the work.

Response- Thank you for your comments. The degree of novelty and originality of the work have been refreshed. (Highlighted in yellow colors)

- Are the results obtained in triplicate? Include more statistical information. Please include a subchapter for statistical analysis!

Response- Thank you for your comments. The subchapter for statistical analysis has been added in the text. (Highlighted in yellow colors)

I didn’t see the citations 6-10 in the text.

Response- Thank you for your comments. The original citations 6-10 have changed to 16-20 and have cited in the text.

-The anti-inflammatory potential needs to be compared with other compounds. The scientific dialogue is quite weak.

Response- Thank you for your comments. The anti-inflammatory part have been refreshed in the text.

- Include in the text potential research directions. What are the future applications? What are the next research directions?

Response- Thank you for your suggestions. The point had been included in introduction and conclusions.

Reviewer 2 Report

The authors considered recent advances in a investigation and chromatographic separation of extracts from the roots of Cimicifuga taiwanensis. Moreover, several modern new compounds are also isolated. It should be of some interest to the readers in the field of medicine and relevant applications. However, the content is not well organized, a major revision is needed for final publication.
(1) in the introduction and conclusion sections, the merit of using Cimicifuga substracts is not well explained. Also, the disadvantages of using this bioactive metabolites in other types of activity should be claimed.
(2) It is important to structure the information not only in the 1H, 13C NMR table method, but also in other analysis methods too. The Tables should be more comprehensive for the all methods.
(3) It is necessary to expand the comment on Biological Studies. Does the control involve comparison with known analogues? What are the advantages, what are the disadvantages?
(4) Please add refer to a high-quality reviews: 
(a) Jain C., Khatana S., Vijayvergia R. Bioactivity of secondary metabolites of various plants: a review //International Journal of Pharmaceutical Sciences and Research. – 2019. – 10 (2): 494-498.
(b) Pan R. et al. Exploring structural diversity of microbe secondary metabolites using OSMAC strategy: a literature review //Frontiers in microbiology. – 2019. – 10:294
(c) Tiwari R., Rana C. S. Plant secondary metabolites: a review //International Journal of Engineering Research and General Science. – 2015 – 3(5):661-670.
(5) IF the authors would not find a better way to reorgainze the content with more informative figures and tables. It should be rejected.

Author Response

Reviewer 2

The authors considered recent advances in a investigation and chromatographic separation of extracts from the roots of Cimicifuga taiwanensis. Moreover, several modern new compounds are also isolated. It should be of some interest to the readers in the field of medicine and relevant applications. However, the content is not well organized, a major revision is needed for final publication.
(1) in the introduction and conclusion sections, the merit of using Cimicifuga substracts is not well explained. Also, the disadvantages of using this bioactive metabolites in other types of activity should be claimed.

Response- Thank you for your comments. The point had been included in introduction and conclusion.

(2) It is important to structure the information not only in the 1H, 13C NMR table method, but also in other analysis methods too. The Tables should be more comprehensive for the all methods.

Response- Thank you for your comments. The Table has been refreshed according to your suggestions.

(3) It is necessary to expand the comment on Biological Studies. Does the control involve comparison with known analogues? What are the advantages, what are the disadvantages?

Response- This part has been revised again, and the constituents of the same genus with anti-inflammatory activity in the past are compared with this study.

(4) Please add refer to a high-quality reviews:

(a) Jain C., Khatana S., Vijayvergia R. Bioactivity of secondary metabolites of various plants: a review //International Journal of Pharmaceutical Sciences and Research. – 2019. – 10 (2): 494-498.
(b) Pan R. et al. Exploring structural diversity of microbe secondary metabolites using OSMAC strategy: a literature review //Frontiers in microbiology. – 2019. – 10:294
(c) Tiwari R., Rana C. S. Plant secondary metabolites: a review //International Journal of Engineering Research and General Science. – 2015 – 3(5):661-670.
(5) IF the authors would not find a better way to reorgainze the content with more informative figures and tables. It should be rejected.

Response- The review (a) has been referred according to your suggestions.

Reviewer 3 Report

Dear authors,

the manuscript deals with the analysis of the compounds found in the Cimicifuga taiwanensis.

The manuscript contains different analytical techniques used for the structural characterization.

Although the information may be interesting for the readers and possibly an interesting new finding, the reader is just confronted with analytical results.

  1. The introduction needs a deeper presentation of the whole subject. The sentences are few and repeated twice.
  2. The reader must be informed, in what cases this plant is useful, what exactly is treated, in what is it successful, why we may need this plant and its constituents, what type of interactions happen with these molecules ?
  3. What are any side effects, if available
  4. Was there any substantial study about this plant ?
  5. What is the anti-inflammatory effect expected exactly ?
  6. The references need to be updated as well. Most of them are quite old.
  7. In the conclusions, the reader must clearly have a bottom line of understanding, what are the uses, potential uses and future applications, especially in the mentioned anti-inflammatory effects.

Author Response

  1. The introduction needs a deeper presentation of the whole subject. The sentences are few and repeated twice.

Response- Thank you for your comments. The Introduction section has been refreshed according to your suggestions.

  1. The reader must be informed, in what cases this plant is useful, what exactly is treated, in what is it successful, why we may need this plant and its constituents, what type of interactions happen with these molecules ?

Response- Thank you for your comments. The point has been included in Introduction.

  1. What are any side effects, if available

Response- Thank you for your comments. The research plant has not been conduct previously.

  1. Was there any substantial study about this plant ?

Response- Thank you for your comments. It has been used as anti-inflammatory, and antipyretic agents in Chinese medicine.

  1. What is the anti-inflammatory effect expected exactly ?

Response- Thank you for your comments. In the pharmacological anti-inflammatory experiments, the active compounds belong to the skeleton of 9,10-seco-cimigenol-type triterpene with an ether bond between C-3 and C-10, which is different from previous studies on the same genus in the literature. This finding merits our continued further pharmacological mechanism of the active constituents.

  1. The references need to be updated as well. Most of them are quite old.

Response- Thank you for your comments. The references have been updated as well.

  1. In the conclusions, the reader must clearly have a bottom line of understanding, what are the uses, potential uses and future applications, especially in the mentioned anti-inflammatory effects.

Response- Thank you for your comments. The conclusions has been revised according to your suggestions.

Reviewer 4 Report

This paper describes the structural elucidation of 6 compounds extracted from C. taiwanensis as well as their biological activity (inhibitory effects of compounds on LPS-activated NO productions in macrophages).

Globally:

- The structure of the introduction may need to be reviewed. The first part concerns the description of different species of Cimicifuga present in different countries with a description of their biological activity. The 2nd paragraph focuses on C. taiwanensis. But is this plant already used in traditional medicine? Why are authors interested in it? It is difficult to see the relationship between the two paragraphs.

- The “Results and discussion” paragraph is a “Results” paragraph. There is no discussion (except for the elucidation of the structures). The authors do not compare their results obtained for C taiwanensis (chemical composition and biological activity) with those of the other species of the genus Cimicifuga described in the introduction.

More precisely:

- Why for certain molecules the authors determined the melting point (3), the αD (3,4 and 6) and not for the others?

- Did the authors make a UV spectrum for all the molecules?

- Paragraph 2.2 is not very well written. Authors should write sentences rather than indicate a) b)…

- Paragraph 3: the authors do not provide any information on the operating conditions for obtaining the spectra of all techniques (UV, IR, NMR, MS)

Author Response

Review 4

This paper describes the structural elucidation of 6 compounds extracted from C. taiwanensis as well as their biological activity (inhibitory effects of compounds on LPS-activated NO productions in macrophages).

Globally

- The structure of the introduction may need to be reviewed. The first part concerns the description of different species of Cimicifuga present in different countries with a description of their biological activity. The 2nd paragraph focuses on C. taiwanensis. But is this plant already used in traditional medicine? Why are authors interested in it? It is difficult to see the relationship between the two paragraphs.

Response- Thank you for your comments. The introduction part has been revised according to your suggestions.

- The “Results and discussion” paragraph is a “Results” paragraph. There is no discussion (except for the elucidation of the structures). The authors do not compare their results obtained for C taiwanensis (chemical composition and biological activity) with those of the other species of the genus Cimicifuga described in the introduction.

Response- Thank you for your suggestions. The Results and discussion has been revised according to your suggestions.

More precisely:

- Why for certain molecules the authors determined the melting point (3), the αD (3,4 and 6) and not for the others?

Response- Thank you for your suggestions. Except for oily substances, no need to determine the melting point. In addition, all isolated 1-6 have chiral center in molecules and all compounds have determined their [α]26 D.

- Did the authors make a UV spectrum for all the molecules?

Response- No. We made the UV spectrum for the compounds 3, 5 & 6 which contain conjugated double bond.

The Results and discussion has been revised according to your suggestions.

Response- Thank you for your suggestions. The Results and discussion has been revised according to your suggestions.

- Paragraph 2.2 is not very well written. Authors should write sentences rather than indicate a) b)…

Response- Thank you for your suggestions. The Paragraph 2.2 has been revised according to your suggestions.

- Paragraph 3: the authors do not provide any information on the operating conditions for obtaining the spectra of all techniques (UV, IR, NMR, MS)

Response- Thank you for your suggestions. The information on the operating conditions for obtaining the spectra of all techniques has been included according to your suggestions.

Round 2

Reviewer 1 Report

The authors responded appropriately. The article may be considered for publication.

Author Response

Thank you for your affirmation and positive comments.

Reviewer 3 Report

Dear authors,

you made a great difference to the first version. The reader gets some background information and the topic is more understood. Your findings are very interesting and have great potential. The work is done neatly.

Only few things remain:

1-Please change wording and edit the information given in the sentences on line 406 to 410. You may also explain, what you mean with "yang toxin", so it can be better understood.

2-The new sentences need to be checked for small mistakes in grammar in general example line 429.

3-The provided references are :

-some were updated to newer publications

-others remained the same

-new ones were added. The new ones mainly are old references. My advice is to replace the old reference with new ones.

4-The General Experimental Procedures are complete but lack any information about the detailed analysis procedures.

After these changes the manuscript can be accepted.

Wish you the best 

Author Response

1-Please change wording and edit the information given in the sentences on line 406 to 410. You may also explain, what you mean with "yang toxin", so it can be better understood.

Response- Thank you for your comments. The point has been revised again.

2-The new sentences need to be checked for small mistakes in grammar in general example line 429.

Response- Thank you for your comments. The sentence has been revised as “The active structures obtained this time are ……”

3-The provided references are :

-some were updated to newer publications

-others remained the same

-new ones were added. The new ones mainly are old references. My advice is to replace the old reference with new ones.

Response- Thank you for your comments. The new references have replaced old ones.

4-The General Experimental Procedures are complete but lack any information about the detailed analysis procedures.

Response- Thank you for your suggestions. The information about the detailed analysis procedures had been refreshed.

Reviewer 4 Report

The authors answered well to the reviewer suggestions.

And they compare their results with those described in the literature for other genera of Cimicifuga.

However, could they expand on the comparison they make (lines 298-312) between the molecules they identified in C. taiwanensis and those described in the literature for other genera of Cimicifuga? They compare the biological activities but do not correlate  (or a little) the structures with the activity: "cycloartane" (line 301) (litterature) versus “9,10-seco-cimigenol-type triterpene, hydrophobic groups, such as carbonyl and acetoxy, instead of a hydroxyl group at C-3, C-11 or C-25 are essential for anti-inflammatory activity.” (lines 295-296) (their results).

They should provide more details on the structure of the compounds described in the literature as having anti-inflammatory properties and they should correlate their structure with their activity versus their results obtained with C. taiwanensis.

Author Response

Response- Thank you for your comments. The information has been revised and included in the text.